

# Comparing audio- and video-delivered instructions in dispatcher-assisted cardiopulmonary resuscitation with drone-delivered automatic external defibrillator: a mixed methods simulation study

Hyun-Jung Kim[1], Jin-Hwa Kim[2] and Dahye Park[3]

[1] Department of Nursing, Daewon University College, Jecheon, Chungbuk, Republic of Korea
[2] Department of Emergency Medical Technology, Daewon University College, Jecheon, Chungbuk, Republic of Korea
[3] Department of Nursing, Semyung University, Jecheon, Chungbuk, South Korea

Corresponding author
Dahye Park, dhpark@semyung.ac.kr

## ABSTRACT

This study compared first responders' cardiopulmonary resuscitation (CPR) performance when a dispatcher provides audio instructions only and when both audio and video instructions are given. In the simulation, an automatic external defibrillator (AED) was delivered via drone in response to a cardiac arrest occurring outside a hospital setting. Participants' qualitative experiences were also explored. An exploratory sequential mixed methods design was used. AEDs were delivered to college students via drone with one group receiving both audio and video instructions and the other receiving audio-only instruction, and differences in CPR performance and accuracy were compared. After completion, focus group interview data were collected and analyzed. Video-based instruction was found to be more effective in the number of chest compressions ($p < 0.01$), chest compression rate ($p < 0.01$), and chest compression interruptions ($p < 0.01$). The accuracy of the video group for the chest compression region was high ($p = 0.05$). Participants' experiences were divided into three categories: "unfamiliar but beneficial experience," "met helper during a desperate and embarrassing situation," and "diverse views on drone use." Our results lay the groundwork for a development plan for providing emergency medical services using drones, as well as the preparation of guidelines for dispatchers on the provision of video instructions.

## INTRODUCTION

Despite advances in medical technology and improvements in economic standards, the prevalence of heart disease and resulting mortality rates are rapidly increasing in Korea due to the aging social structure and Westernized eating habits. Rapid and appropriate

cardiopulmonary resuscitation (CPR) can improve spontaneous circulation recovery and survival in patients experiencing cardiac arrest (*Lu, Fang & Lin, 2019*). Brain damage occurs 4 min after the onset of cardiac arrest, and after 10 min, the person will suffer irreversible brain damage or brain death. Therefore, immediate administration of first aid is critical in determining whether the patient is resuscitated (*Perkins et al., 2015*). In addition, the mortality rate increases by 7–10% every minute a patient experiences cardiac arrest rhythms that require defibrillation. Therefore, shortening the time from cardiac arrest to defibrillation is an important factor in patient survival (*Hallstrom, Ornato & Weisfeldt, 2004*).

The average arrival time of ambulances in Korea is 9 min (*Jang et al., 2016*). To improve patient survival rates and neurological prognosis in cardiac arrest, there is a need for medical services that can rapidly deliver automatic external defibrillator (AED) devices.

A geographic information system model that can be used to deliver drones to patients experiencing cardiac arrest outside of a hospital setting was recently proposed (*Boutilier et al., 2017*; *Pulver, Wei & Mann, 2016*; *Ringh et al., 2015*). Such a system is less geographically and spatially limited than traditional ambulances, and the operating costs are low. The time required to deliver an AED via drone was 19 min less than the ambulance arrival time in a recent study (*Claesson et al., 2016*). In a follow-up study (*Claesson et al., 2017*), AED delivery using a drone took 5 min, 21 s, 16 min less than an ambulance to the same site (22 min). However, to the best of our knowledge, no studies have yet used drone-delivered AEDs in actual situations in Korea. Therefore, there is a need for research on drones that can overcome topographic conditions and quickly deliver defibrillators to the site of cardiac arrest.

It is not only important to provide medical services promptly in the event of cardiac arrest, but also to provide the services accurately. Accordingly, to increase the rate of CPR during cardiac arrest, it was performed based on the dispatcher's instructions. In the event of an out-of-hospital cardiac arrest, dispatchers provide guidance to callers, helping them confirm cardiac arrest and initiate CPR until emergency rescuers arrive (*Lee et al., 2020*). Currently, dispatchers provide verbal CPR guidance to bystanders via mobile phone. However, it is difficult to confirm whether the CPR performer correctly understands and follows the audio instructions of the dispatcher (*Williams et al., 2006*). To address this problem, a mobile phone with a two-way video call function can be used to provide both audio and video instructions, allowing dispatchers to check whether CPR is being properly performed in real time, and to correct errors immediately (*Lin et al., 2017*). However, this multimedia approach has not been generalized, and there is a lack of research comparing the CPR performance and accuracy of bystanders following audio and video instructions. Therefore, this study compared the CPR performance and accuracy of audio and video instructions for CPR performers when an AED is delivered via drone to the site of a cardiac arrest outside a hospital, and explored the experience provided by the audio- and video-instruction CPR using drones and how it affected the CPR process.

## MATERIALS & METHODS

### Design

This study followed an exploratory sequential mixed method design. For the quantitative component, AEDs were delivered to college students by drones and CPR instructions were delivered by a dispatcher. A nonequivalent control group posttest only design was conducted between the group that received audio-only instructions and the group that received mixed video and audio instructions, to compare differences in CPR performance and accuracy. For qualitative research, after the participants' experiments were completed, data were collected and analyzed through focus group interviews. The study was conducted according to the guidelines of the Declaration of Helsinki, and approved by the Ethics Committee of the University of Semyung (protocol code SMU-2020-07-003 and date of approval August 7, 2020). Written informed consent has been obtained from all participants for their data to be used for scientific research.

### Instruments

The drone used to transport the AED was equipped with a satellite positioning system (GPS) and high-definition camera (Fig. 1; maximum load: 16 kg, maximum cruising speed: 10 m/s, maximum altitude: 2,000 m, JW002).

The CPR performance evaluation was performed using Resusci Anne®, QCPR® manikin (Laerdal, Norway) and a CPR quality assessment program connected to the manikin (Laptop PC, Laerdal, Norway with Sim Pad Skill Reporter and Resusci Anne® Wireless Skill Reporter software installed; Fig. 2). The scores were automatically recorded and saved by the Sim Pad Skill Reporter. Three evaluators were present to assess the accuracy of participants' movements and sequences that could not be evaluated by the Sim Pad Skill Reporter. Performance was judged based on the following criteria: consciousness confirmation (2 points), help request (2 points), breathing confirmation (4 points), airway management (4 points), artificial respiration (24 points: 8 points ×3 cycles), chest compression (24 points: 8 points ×3 cycles), and automatic defibrillator use (10 points). Users were given 0 points for a poor performance, 1 point for average, and 2 points for good. The total score of the three evaluators was calculated as the average score, and the time taken from the arrival of the AED to the execution of the defibrillation was measured using a stopwatch. The AED was a fully automatic Laerdal® AED Trainer 2 (Medtronics, Physiocontrol, USA), as shown in Fig. 3.

The focus group instrument consisted of three main types of questions. The first, introductory questions, explored basic personal information (e.g., whether participants had completed basic life support-provider (BLS-P) training, or had actual CPR experience). Key questions asked about participants' experiences of the drones and audio/video CPR instructions (e.g., "How was the experience of performing CPR with the AED delivered by drone?" and "What was the experience like for dispatchers giving audio and video CPR instructions?") Closing questions provided space for additional statements and thoughts related to the drones and performing CPR following the audio/video instructions.

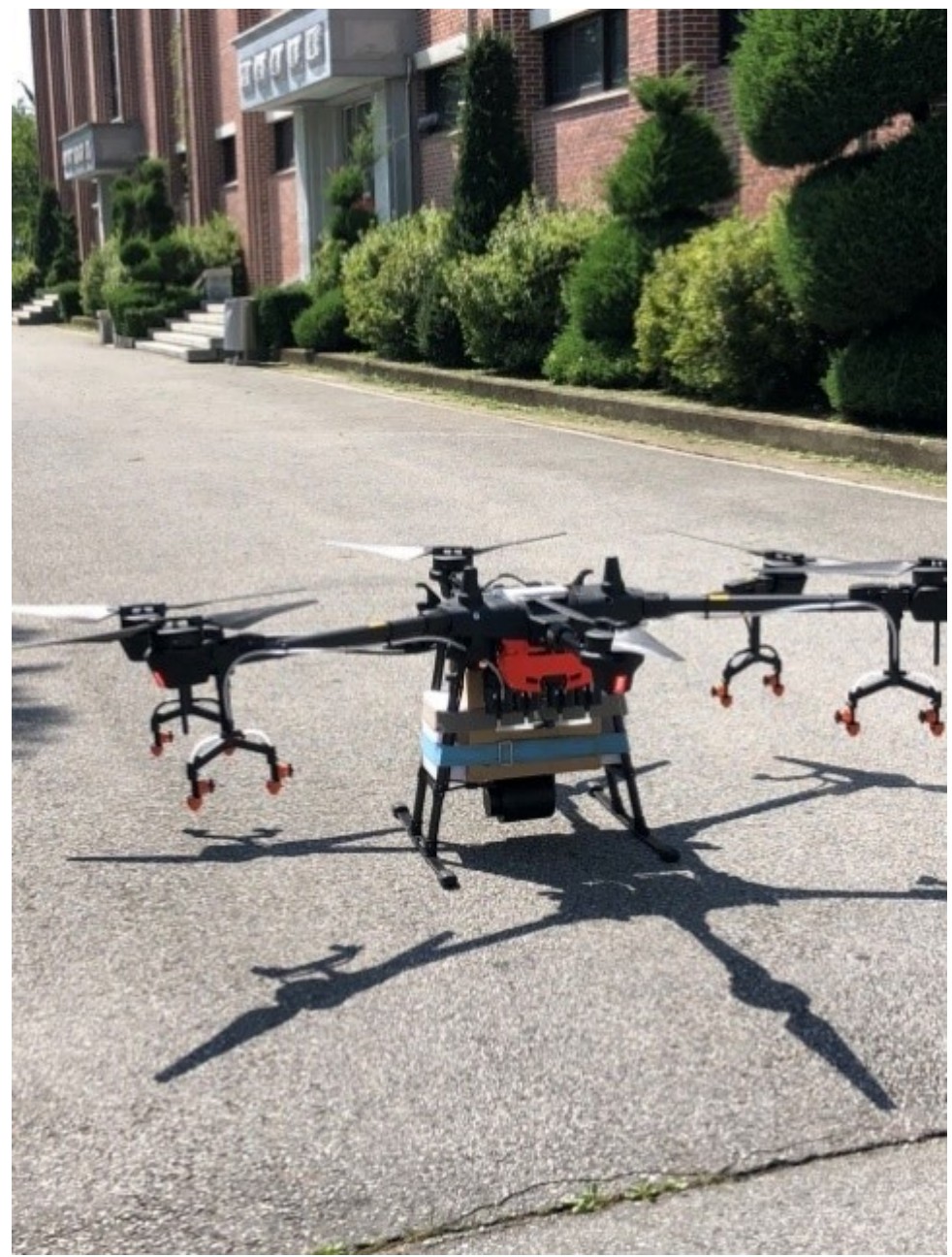

**Figure 1  The drone used to transport the AED.**

## Setting

The experiment was conducted in a location with adequate lighting and where external noise was blocked. Consenting participants, who were eligible to participate in the study, were randomly allocated to either the group receiving audio-only instructions or the group receiving mixed video-audio instructions. Group randomization was based on a computer-generated sequence list. The randomized list was prepared by an independent research

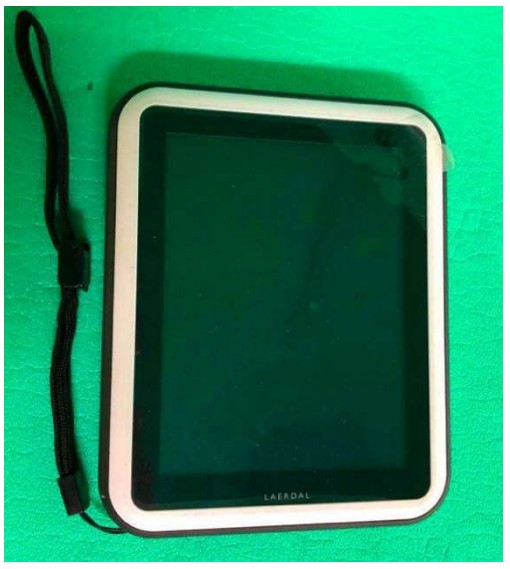 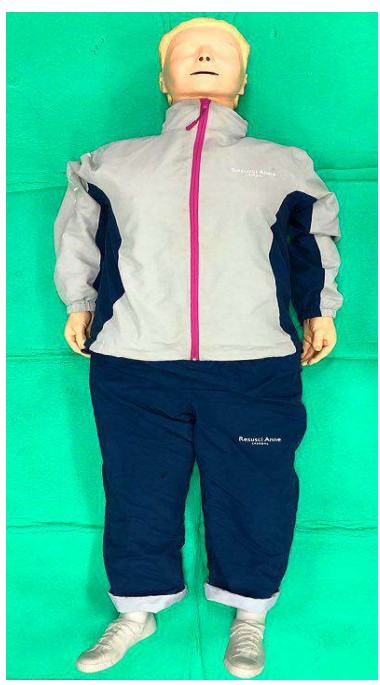

**Figure 2  SimPad and Resusci Anne Q-CPR.**

assistant who was not involved in recruiting participants and concealed in sequentially numbered, sealed, opaque envelopes. Participants acted alone as the first bystander and rescuer in a simulated situation where cardiac arrest was suspected. Participants were notified on the day of the experiment that an automatic defibrillator would be delivered using a drone. The participants performed CPR by calling the emergency medical system (EMS, 119: virtual number) at the simulation site and receiving instructions (audio/video) from the dispatcher, as shown in Fig. 4.

Three CPR motion evaluators and one dispatcher were standardized through pre-evaluation and training, each with over 10 years' experience as a first-class emergency medical technician or BLS-P-Instructor for the American Heart Association. The motion evaluators rated the performance of the control and experimental groups. The evaluators could not hear the dispatcher's instructions, which allowed them to judge the CPR performances based on motions only.

The guidance programed on the virtual 119 phone call explained the CPR algorithm (video/audio guidance) and informed the callers that ambulances and drones were being deployed. Based on a previous study (Claesson et al., 2017), the simulated flight time of the drone was set to 5 min. After CPR had been performed for 5 min, it was announced that an automatic defibrillator was delivered to the ground using a drone at a point 50 m from the incident site. To provide visual feedback, the drone was marked with a red AED bag at an altitude of 10 m. The simulation ended when the participant attached the electrodes of the AED to the manikin.
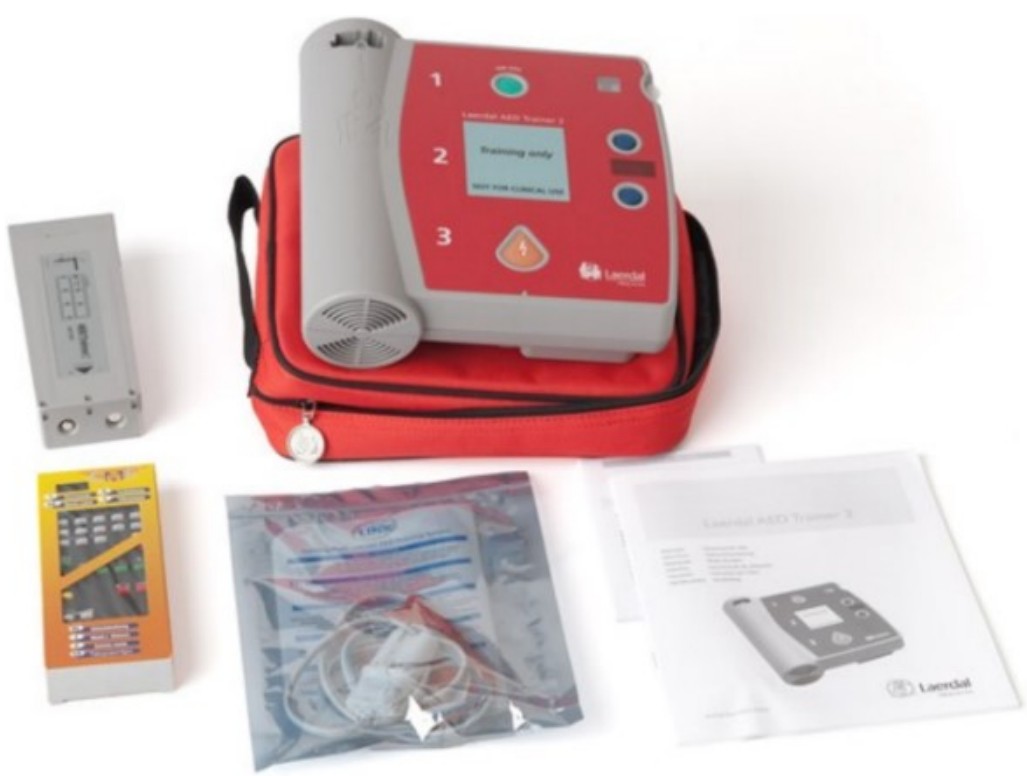

**Figure 3**  **The automatic external defibrillator (AED) device.**

## Audio instruction (control group) method and video instruction (experimental group) method

The dispatcher conducted the training in an office isolated from the participants. The audio instruction (control) was performed flexibly so that it could be modified according to the participant's understanding and performance using a basic mobile phone. The dispatcher wore a Bluetooth headset (SHM-612) capable of verbally transmitting and receiving CPR instructions. Video instructions (experimental group) were performed using a video mobile phone. The phone had a 12-megapixel camera with a resolution of 2220 × 1080 capable of two-way video calls, and a Galaxy S8+, SM-G955U1 model with an HSDPA 150 Mbps (upload transmission rate). The dispatcher, while assessing the participant's CPR performance through the audio and video, provided instructions, according to the American Heart Association guidelines, on things such as CPR sequence and posture correction.

## Data collection

In *Sanfridsson et al.*'s (*2019*) mixed study, 16 people were tested and interviewed. Moreover, for content analysis, a qualitative research method, interview data from five to 25 people is considered appropriate (*Creswell & Plano Clark, 2007*). In this study, there were 24 participants (assuming a 20% dropout rate).
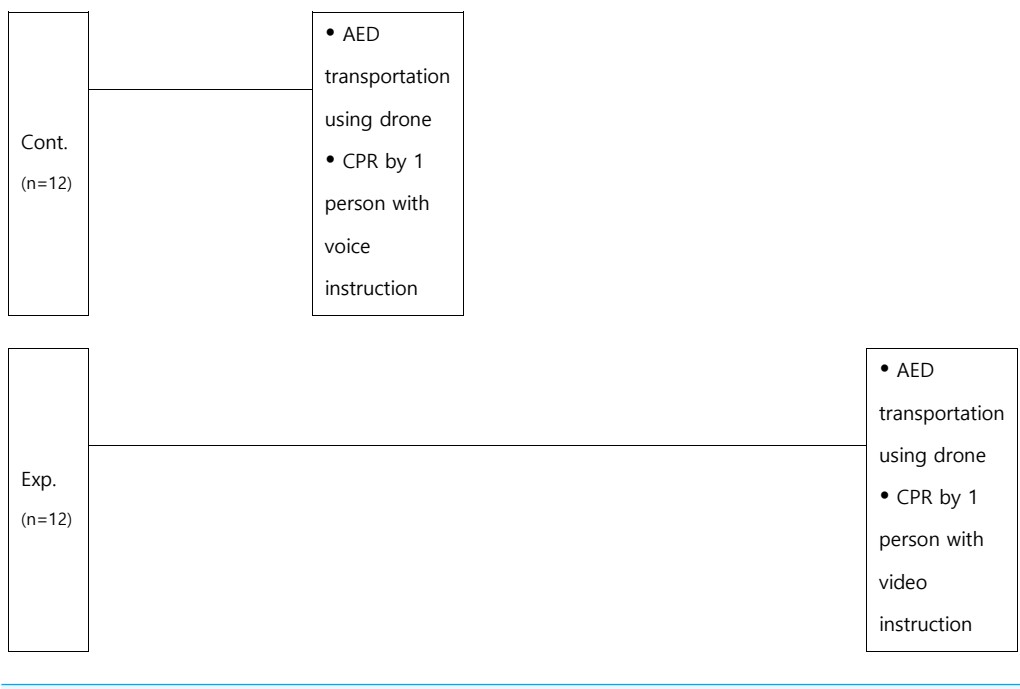

**Figure 4 Flowchart of simulations.**

To minimize the difference between the experiments in terms of collecting quantitative data, no advance education was provided to the participants. To minimize contamination and errors that could affect the results via the exchange of information about the experiment or acquisition of prior knowledge between participants, the control group (audio instruction CPR) and the experiment group (video instruction CPR) were assigned at unspecified times. Subsequently, the experiment was conducted individually.

To collect qualitative data, focus group interviews were conducted with the 24 participants. Semi-structured interviews were conducted in groups of four to six participants, using open-ended questions. The participants agreed to participate in the recorded interviews, which were conducted in a quiet and comfortable seminar room.

## Data analysis
### Quantitative data analysis
The data were analyzed using SPSS Windows 25.0. First, participants' general characteristics were analyzed using real numbers and percentages, means, and standard deviations. Next, the Kolmogorov–Smirnov test was performed to verify the normality of the participant group, and then a non-normal distribution was verified using the Wilcoxon signed-rank test, a nonparametric statistical method.

### Qualitative data analysis
In the post-experiment interview, open questions were used; participants' responses were recorded and transcribed verbatim. The analyzed results were made available to participants, so they could verify that their statements were transcribed without distortion.

**Table 1  General characteristics of participants ($N = 24$).**

| | Audio-instructed CPR N(%) | Video-instructed CPR N(%) | $X^2$/t(p) |
|---|---|---|---|
| Gender | | | |
| Men | 7(58.3) | 4(33.3) | 1.510(.219) |
| Women | 5(29.2) | 8(66.7) | |
| Age | 22.33 ± 1.67 | 22.42 ± 1.98 | -.112(.811) |
| Height | 165.58 ± 5.00 | 164.50 ± 7.69 | 2.299(.099) |
| Weight | 64.75 ± 11.06 | 61.00 ± 13.67 | .739(.310) |

To secure transferability, focus group interviews were conducted in four groups, and data were collected up to the saturation state where no new content was yielded. *Elo & Kyngäs (2007)* content analysis method was used to analyze the focus group interview data. As the material obtained through the interviews was repeatedly read, related words or phrases were underlined and annotated in the margins. As the responses were read several times, the sentences or phrases judged to be meaningful were underlined. Meaningful statements were then categorized and described as universal and abstract concepts. Finally, the organized categories were comprehensively described so that the phenomena experienced were well revealed from the participants' point of view. The researchers reviewed previous studies on related content and continuously checked to ensure that their subjectivity did not intervene in the analysis process. For the analyzed data, the peer examination was conducted by one professor in the department of nursing with extensive experience in qualitative research and one researcher who completed a qualitative research methodology class in graduate school.

## RESULTS

### General characteristics of participants
Twelve participants received verbal CPR instructions, and 12 received CPR instructions over video. There was no significant difference in gender, age, height, and weight between the two groups, and thus, homogeneity was ensured (Table 1).

### CPR performance
The number of chest compressions was 96.92 ± 12.33 in the audio instruction CPR group, and 121.42 ± 17.23 in the video instruction CPR group ($p < 0.01$). The rate of chest compression was 47.08 ± 7.79% in the audio instruction CPR group, and 62.67 ± 12.75% in the video instruction group ($p < 0.01$). The hands-off time during chest compressions was 17.08 ± 2.71 s in the audio instruction CPR group, and 10.83 ± 2.92 s in the video instruction CPR group ($p < 0.001$). There were significant differences between the two groups in the number of chest compressions, rate of compressions, and stopping time.

In artificial respiration, the total respiration rate was 3.50 ± 2.20 times in the audio instruction CPR group and 5.50 ± 2.75 times in the video instruction CPR group, showing a significant difference between the two groups ($p = 0.05$). The total score for CPR performance was 19.25 ± 11.85 points for the audio instruction CPR group, and

**Table 2  Differences in CPR performance based on instruction method ($N = 24$).**

| | | Audio instruction | Video instruction | P-value[a] |
|---|---|---|---|---|
| Chest compressions | Score (%) | 76.67 ± 32.55 | 86.58 ± 16.09 | .621 |
| | Frequency | 96.92 ± 12.33 | 121.42 ± 17.23 | .002 |
| | Average depth (mm) | 52.17 ± 12.10 | 53.67 ± 5.30 | .487 |
| | Relaxation rate (%) | 71.67 ± 34.55 | 90.83 ± 16.39 | .063 |
| | Implementation rate (%) | 47.08 ± 7.79 | 62.67 ± 12.75 | .001 |
| | Hand position accuracy (5) | 86.25 ± 28.56 | 98.42 ± 5.49 | .074 |
| | Compression speed (times/min) | 108.58 ± 3.60 | 104.50 ± 7.19 | .099 |
| | Hands-off time | 17.08 ± 2.71 | 10.83 ± 2.92 | .000 |
| Artificial respiration | Score (%) | 57.42 ± 37.59 | 73.58 ± 33.04 | .505 |
| | Total respiration (times) | 3.50 ± 2.20 | 5.50 ± 2.75 | .048 |
| | Total average amount (ml) | 370.75 ± 193.67 | 430.50 ± 147.10 | .435 |
| | Breathing at the appropriate volume (%) | 57.75 ± 45.63 | 72.42 ± 31.03 | .929 |
| | Speed (times/min) | 1.58 ± 1.17 | 5.58 ± 1.44 | .081 |
| Total score | | 19.25 ± 11.85 | 63.17 ± 24.10 | .000 |

**Notes.**
[a]Statistical significance was measured by the Mann–Whitney $U$-test.

63.17 ± 24.10 points for the video instruction CPR group, indicating that overall CPR performance was significantly higher in the audio instruction CPR group ($p < 0.001$; Table 2).

## CPR accuracy

The Cronbach's alpha of the total CPR accuracy score by the three evaluators was 0.91. The evaluation score for consciousness confirmation was 1.50 ± 0.52 in the audio instruction CPR group and 2.00 ± 0.00 in the video instruction CPR group ($p < 0.01$); 119 reports and defibrillator requests were 0.92 ± 0.28 in the audio instruction CPR group and 1.42 ± 0.51% in the video instruction CPR group ($p = 0.01$). The evaluation score for the chest compressions, which measured the location of the chest compression, the shape of the performers' hands, and their body posture, was 23.08 ± 1.38 in the audio instruction CPR group and 23.92 ± 0.29 in the video instruction CPR group ($p = 0.05$). The score for 1st cycle chest compression was 7.69 ± 0.45 in the audio instruction CPR group and 8.00 ± 0.00 in the video instruction CPR group, respectively ($p = 0.01$). The results showed that the accuracy of CPR, as gauged by the evaluators, was significantly higher in the video instruction CPR group in terms of consciousness check, 119 report and defibrillator request, and chest compressions compared to the audio instruction CPR group (Table 3).

## Qualitative content analysis results

From the content analysis of the participants' post-experiment responses, six subcategories were extracted, from which three main categories were derived (Table 4).

### *Unfamiliar but beneficial experience*

This category includes the subcategories of "regret for poor performance" and "vibrancy, as if experiencing a real situation." While participating in the experiment, some participants

**Table 3 Difference in evaluation score based on instruction method (N = 24).**

|  | Audio instruction | Video instruction | P-value [1] |
|---|---|---|---|
| Consciousness check | 1.50 ± 0.52 | 2.00 ± 0.00 | .006 |
| 119 report & defibrillator request | 0.92 ± 0.28 | 1.42 ± 0.51 | .010 |
| Breathing check (4) | 3.58 ± 0.51 | 7.33 ± 1.15 | .868 |
| Chest compressions (8) average | 7.69 ± 0.45 | 7.97 ± 0.09 | .050 |
| 1 cycle | 7.69 ± 0.45 | 8.00 ±.00 | .014 |
| 2 cycle | 7.75 ± 0.62 | 8.00 ± 0.00 | .149 |
| 3 cycle | 7.83 ± 3.90 | 7.91 ± 0.289 | .546 |
| Total score | 23.08 ± 1.38 | 23.92 ± 0.29 | .050 |
| Airway check (4) | 4.00 ± 0.00 | 3.83 ± 0.57 | .317 |
| Artificial respiration (8) average | 6.05 ± 1.20 | 6.58 ± 1.27 | .257 |
| 1 cycle | 6.58 ± 0.90 | 6.50 ± 0.80 | .452 |
| 2 cycle | 6.91 ± 1.08 | 7.00 ± 1.21 | .738 |
| 3 cycle | 4.66 ± 3.52 | 6.25 ± 2.26 | .327 |
| Total score | 18.17 ± 3.61 | 19.75 ± 3.82 | .257 |
| Defibrillator (10) | 9.00 ±.85 | 9.00 ± 1.04 | .707 |
| Defibrillation time | 31.64 ± 6.04 | 32.23 ± 7.90 | .817 |
| Total | 61.75 ± 3.91 | 64.92 ± 5.13 | .072 |

**Table 4 Results of the qualitative content analysis (N = 24).**

| Category | Subcategory |
|---|---|
| Unfamiliar but beneficial experience | Regret for poor performance |
|  | Vibrancy, as if experiencing a real situation |
| Met helper during a desperate and embarrassing situation | Comfort that you are not alone |
|  | Compensation for insufficiencies |
| Different views on the use of drones | Conflicting feelings about the size of the drone |
|  | Varying views on the accessibility of drones |

expressed some regret and embarrassment caused by the unexpected situation, because it was unfamiliar or not controlled. However, all participants agreed that they had meaningful and beneficial experiences.

*Regret for poor performance.* Participants commented on experiences in which smooth practice was difficult for various reasons. In particular, participants showed remarkable differences in responses to the audio and video instructions depending on whether they had completed BLS-P training in the past. Participants who had not completed this training responded that it was difficult to accurately grasp the depth of chest compressions and how to breathe in; therefore, more detailed explanations were needed.

"It was difficult to judge the correct depth of chest compression and artificial respiration because I did not have an accurate knowledge of CPR." (Participant 7)

On the other hand, some students who had already completed BLS-P training were concerned that intervention would be delayed if they had to wait to be directed by paramedics in a real situation that required a rapid management process.

"I thought that patients need to be treated quickly as in real life, but it was a pity and difficulty because I had to proceed with the paramedics." (Participant 21)

Some participants were also hindered by uncontrolled situational variables. For example, the defibrillator sounds made it difficult to hear the call.

"It was a bit confusing because the defibrillator and the call sound overlapped." (Participant 20)

Furthermore, although the participants did not actually experience it, they were worried about what to do if the connection was lost during the call.

"In the case of audio instruction, there is a risk that the mobile phone connection may not be successful, or the connection may be disconnected due to battery discharge." (Participant 19).

*Vibrancy, as if experiencing a real situation.*  Most participants initially had doubts about the novel experimental method, but expressed their satisfaction, saying that they were immersed in the simulated situation as if they had actually experienced it, and that this experience would help a lot in actual practice in the future.

"I think I had a good experience by performing the simulation well as in the real world. I think this simulation is a good idea." (Participant 1)

### Met helper during a desperate and embarrassing situation

This category includes the subcategories of "comfort that you are not alone" and "compensation for insufficiencies." Most participants relied on the dispatcher and the drone to compensate for their lack of knowledge and insufficient skills; thus, they resolved the situation.

*Comfort that you are not alone.*  Most of the participants said they felt anxious, scared, and lonely, when faced with the burden of solving the situation on their own. The drones and dispatchers that appeared then gave them the comfort that they were not alone. Interestingly, even though the drone is a machine rather than a human, the participants expressed that they thought of the drone as an "assistant" or "I was not lonely because the drone came," expressing that the drone provided them emotional comfort.

"I was frustrated and embarrassed because I was going to do it alone in the situation where no one was there, but it was nice that the drone seemed to act as an assistant." (Participant 22)

*Compensation for insufficiencies.*  In situations where the participants had to perform CPR on their own, not only those who did not complete BLS-P training but also those who completed it felt embarrassed. However, incomplete actions could be corrected through audio and video, so even participants who had not completed BLS-P training could follow it easily.

"It was nice to see the treatment method through the video instruction, so I could correct the insufficient parts." (Participant 9)

### Different views on the use of drones

This category includes the subcategories of "contradictory feelings about drone size" and "varying views on the accessibility of drones." All participants commented on the innovation of drone use, but ultimately had different views on this aspect.

*Conflicting feelings about drone size.* Most participants thought the drone was too big, making it dangerous for both the patient and rescuer during takeoff and landing.

"This is my first experience, but there seems to be a danger to both the requestor and the rescuer when the drone takes off and lands." (Participant 7)

However, one participant was satisfied with the large size of the drone, saying that it looked sturdy.

"It was amazing and impressive that the drone looked so big and sturdy that it could deliver an AED through it." (Participant 14)

*Varying views on the accessibility of drones.* Most participants thought the method of receiving an AED via drone would be useful in environments where it is difficult to access defibrillators, such as in the mountains or in remote areas, and they expected that the quality of CPR would be improved compared to conventional single-person CPR.

"I felt it was really practical. If I met a patient with cardiac arrest while hiking, I never thought of what a drone would bring in an environment without an AED." (Participant 22)

However, the participants highlighted some points for improvement. Some judged it not to be effective in this case because large drones could not access narrow places with lots of grass or leaves. In addition, many participants found that it was difficult to use the drone on uneven terrain because it is difficult for the drone to land on a slope. As the drone is so large, the AED landing location may be some distance away from the patient, considering the length of the wings.

"When the AED arrived, it was so far away from the patient that it was unfortunate that there was a gap in the middle. I thought it would be better if the AED could come slightly closer to the patient." (Participant 1)

## DISCUSSION

This study compared the difference in CPR performance and the accuracy of the first responder based on whether audio or video instructions were given by a dispatcher during a cardiac arrest outside a hospital setting. We also attempted to provide basic data that can be used to increase the effectiveness of out-of-hospital CPR by confirming the experience of audio and video CPR instruction and exploring how CPR performance is affected when a defibrillator is delivered using a drone.

To increase the survival rate of patients with cardiac arrest, it is important for first responders to use high-quality CPR and rapid defibrillation. For high-quality CPR, chest compressions should be performed at a rate of 100 to 120 times/min, chest compressions at a depth of five cm or more, and the interruption time of chest compressions should be minimized (within 10 s) (*Perkins et al., 2015*). First, regarding difference in CPR

performance based on instruction delivery method in this study, the number and rate of chest compressions were higher in the video instruction group than in the audio instruction group, and the interruption time was shorter in the video instruction group than the audio instruction group. Though these results are consistent with that of *Lin et al.*'s (*2017*) study, thereby, confirming that the quality of CPR provided by the first responders based on audio and video instructions, the chest compression rate was higher in the video instruction group. Additionally, *Ho et al.*'s (*2016*) study found that when CPR was performed by a first responder based on the dispatcher's audio instructions the CPR initiation rate was lower and CPR delayed, which is similar to *Dong et al.*'s (*2020*) study which found that a smartphone application providing real-time monitoring and audiovisual feedback consistently maintained a higher rate compared to voice-directed CPR. Although these studies proved that video instruction is more useful in cardiac arrest situations, in Korea, audio instructions from the dispatcher are more widely used to help first responders in cardiac arrest situations outside a hospital. It is necessary to expand CPR guidance through video instructions from dispatchers to improve the quality of CPR in Korea, where the IT industry is developing, and mobile phones are widely used. A study on first responders' experiences with the delivery of AEDs via drones to the elderly showed some difficulties in using cell phones (*Sanfridsson et al., 2019*), attaching electrodes, and placing electrodes. Since the age range of first responders may vary, the image instruction method should therefore be included in the educational content of CPR to ensure it can be performed while receiving image-based instruction. Another condition for high-quality CPR is to relax enough to compress the chest. In this study, the video instruction group had a higher chest compression relaxation rate than the audio instruction group, although the difference was not statistically significant. Similar results were found in a study comparing video-based CPR education with traditional CPR education (*Chien et al., 2020*). Therefore, when the dispatcher provides CPR instruction, guidelines should be prepared to emphasize sufficient chest relaxation after chest compression, and the use of feedback devices that can monitor sufficient chest relaxation in CPR education for the public, including medical personnel, should be further activated.

In terms of CPR accuracy, judged by three evaluators, the chest compression score, which measured the location of the chest compression, the shape of the responder's hands, and correct posture, was found to be more accurate in the video instruction group than in the audio instruction group. *Lin et al. (2017)* found audio instructions are limited in that it is not possible to confirm the actions of the first responder; however, with video instruction, problems can be corrected through feedback by confirming the actions of the first responder by audio and video. The accuracy of chest compression by the audio instruction group was high in 1st cycle of chest compressions, but there was no difference from the audio instruction group in the 2nd and 3rd cycles. It is necessary to confirm whether there is no difference between the two groups because of the rescuer becoming fatigued over time. In addition, considering that the average arrival time of 119 paramedics is 9 min, it is necessary to train video dispatchers so that high-quality CPR can be maintained continuously while checking the performance of the rescuer, and a dispatcher video instruction guide should be developed accordingly.

In-depth understanding of the experience of drone use, audio instruction CPR, and video- instruction was further analyzed, and the effect of experience on the CPR process was analyzed. When the drone delivered the automatic defibrillator, three categories of experience were derived: "an unfamiliar but beneficial experience," "met helper during a desperate and embarrassing situation," and "diverse views on drone use." The category of "unfamiliar but beneficial experience" included the subcategories of "regret about poor performance" and vibrancy, as if experiencing a real situation." The experience was unfamiliar because the provision of medical services via drone is not a common practice, and it was judged by the participants that the practical experience of providing the necessary defibrillation for cardiac arrest patients through drones had been provided. The participants received practical experience they would not otherwise have received, and that in itself made it a worthwhile experience for them. In addition, some of the study participants who had obtained the BLS-P qualification were concerned about the delay in intervention in coping with the CPR instructions as they were directed by the dispatcher. This result supports previous findings that it took more time to perform CPR correctly during a simulation and more stress occurred (*Sanfridsson et al., 2019*), than in a real-life situation. Therefore, for the dispatcher and first responder to effectively interact in a cardiac arrest situation, video instruction that can confirm the appropriateness of the rescuer's behavior should be developed. In addition, two sets of guidelines should be developed based on individuals' prior knowledge of CPR. If CPR is possible and the rescuer has experience, minimal intervention is better.

The category of "met helper during a desperate and embarrassing situation" included the subcategories of "comfort that you are not alone" and "compensation for insufficient areas." In a study by *Sanfridsson et al. (2019)*, the interaction with the dispatcher offered participants a sense of relief and dependence, and they could handle emergency situations and perform given tasks more easily. This is because most first responders feel anxious and lonely with the burden of having to solve the problem alone in an emergency situation, but drones bring important defibrillators in cardiac arrest situations. In addition, the dispatcher's instructions on how to properly perform CPR provided comfort to the rescuer that they were not alone, and the dispatcher could compensate for the rescuer's shortcomings.

The category of "diverse views on the use of drone" included the subcategories of "contradicting feelings about drone size" and "varying views on drone accessibility." Most participants thought the drone was too big, making it dangerous during takeoff and landing. However, the results of a previous study found AED delivery by drones to be safe and not difficult (*Sanfridsson et al., 2019*). To reduce the safety concerns and anxiety of rescuers, a safe distance should be secured, and the dispatcher's comments should be included in the audio and video instructions. Most participants said it would be useful to receive a defibrillator via drone in an environment where AEDs are not readily available. They were also positive about the accessibility of the drones, saying that receiving a defibrillator quickly can improve the quality of CPR compared to conventional CPR by a single rescuer. Some participants felt that to be useful in emergency situations in a variety of real environments, the drones must be accessible; the drone in this study was

regarded as too large to be effective in a narrow place or a place with a lot of grass or leaves. Thus, it is necessary to determine appropriate delivery routes according to the region and topography so that the drones can be applied in actual emergency medical services. In addition, a follow-up study using path analysis is needed to understand how defibrillator delivery by drone affects patient survival rate.

This study has several strengths. First, it demonstrated that delivering defibrillators using drones overcomes geographical and temporal limitations. Second, a mixed study, that analyzed the participants' experience by interviewing them, was attempted. Third, it confirmed that video instructions can help ensure that CPR is provided accurately. Despite its strengths, this study also has some limitations. First, since CPR can be performed by anyone regardless of their age but this study's participants were in their 20s, the generalizability the results is limited. Second, since there are the designated area for drones, their take-off and landing were limited in the study.

## CONCLUSION

This study provided evidence that there are differences. in performance, execution time, and accuracy of CPR performed by first responders, according to whether the dispatcher provides audio or video instructions, when an automatic defibrillator is delivered via drone during a cardiac arrest scenario outside a hospital. We also endeavored to gain an in-depth understanding of the experience of drone use and audio and video CPR instruction, and we sought to determine how the experience affected the CPR process. According to this study's results, there were significant differences between the audio- and video-instruction groups, while performing CPR, in the number of chest compressions, rate of chest compressions, and hands-off time during chest compressions. In addition, regarding chest compressions, there were differences between the audio- and video-instruction groups in the location of the chest compressions, the shape of the hand, and the correct posture—which confirm the accuracy of the CPR. Three categories of experience were derived from the participants' experiences: "unfamiliar but beneficial experience," "met helper during a desperate and embarrassing situation," and "diverse views on drone use," and six sub-categories were extracted. These results are significant as they provide basic data that can be used in creating a development plan for emergency medical services using drones and preparing video-instruction guidelines for dispatchers.

To activate medical services using drones, based on the current results, we propose a comparative study on the delivery of defibrillators using drones, and the transportation of defibrillators using ambulances after analyzing the routes appropriate for the region and terrain. Generalizability of the findings are severely limited because there is no system that uses AED-drones, currently. In addition, we propose that a follow-up study confirm the efficacy by developing a dispatcher guide that uses audio and video instructions. Additionally, further studies are required to confirm the effects of audio and video instructions on various age groups and in different areas and terrains where there are no restrictions on drones' takeoff and landing.

### Funding
This paper was supported by Semyung University's University Innovation Support Project in 2020. The funders had no role in study design, data collection and analysis, decision to publish, or preparation of the manuscript.

### Grant Disclosures
The following grant information was disclosed by the authors:
Semyung University's University Innovation Support Project in 2020.

### Competing Interests
The authors declare there are no competing interests.

### Author Contributions
- Hyun-Jung Kim conceived and designed the experiments, performed the experiments, authored or reviewed drafts of the paper, and approved the final draft.
- Jin-Hwa Kim conceived and designed the experiments, performed the experiments, prepared figures and/or tables, and approved the final draft.
- Dahye Park conceived and designed the experiments, performed the experiments, analyzed the data, prepared figures and/or tables, authored or reviewed drafts of the paper, and approved the final draft.

### Human Ethics
The following information was supplied relating to ethical approvals (i.e., approving body and any reference numbers):

The study was conducted according to the guidelines of the Declaration of Helsinki, and ap-proved by the Ethics Committee of the University of Semyung (protocol code SMU-2020-07-003 and date of approval August 7, 2020).

### Data Availability
The raw measurements are available in the Supplemental Files.

### Supplemental Information
Supplemental information for this article can be found online at http://dx.doi.org/10.7717/peerj.11761#supplemental-information.

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
