# Peer review of "Comparing audio- and video-delivered instructions in dispatcher-assisted cardiopulmonary resuscitation with drone-delivered automatic external defibrillator: a mixed methods simulation study"

_PeerJ, doi:10.7717/peerj.11761_

## Round 0.1 · original submission · Major Revisions

The manuscript has been assessed by three reviewers, and they agree with the fact that there are still a few points that need to be addressed. We would be glad to consider a substantial revision of your work, where reviewers' comments will be carefully addressed one by one.

Reviewer 1 ·

Basic reporting

- "Accordingly, to increase the rate of CPR during cardiac arrest, it is performed by the dispatcher.": Sentence is ambiguous. How can the dispatcher do CPR? Pls rephrase
- "pre-hospital cardiac arrest should be "out-of-hospital cardiac arrest"
- "post-mortem experiment" pls rephrase to something less misleading
- Literature review. Suggest the authors incorporate important relevant references including:
1. Lin Y-Y, Chiang W-C, Hsieh M-J, Sun J-T, Chang Y-C, Ma MH-M. Quality of audio-assisted versus video-assisted dispatcher-instructed bystander cardiopulmonary resuscitation: A systematic review and meta-analysis. Resuscitation. 2018 Feb;123:77–85.
2. Ho, A. F. W. et al. Barriers to dispatcher-assisted cardiopulmonary resuscitation in Singapore. Resuscitation 105, 149–155 (2016).
3. Ho, A. F. W. et al. Evaluation of culture-specific popular music as a mental metronome for cardiopulmonary resuscitation: a randomised crossover trial. Proceedings of Singapore Healthcare 28, 159–166 (2019).
4. Dong X, Zhang L, Myklebust H, Birkenes TS, Zheng Z-J. Effect of a real-time feedback smartphone application (TCPRLink) on the quality of telephone-assisted CPR performed by trained laypeople in China: a manikin-based randomised controlled study. BMJ Open. 2020 Oct 1;10(10):e038813.
- there is no discussion on the strengths and limitations of the study

Experimental design

- How were participants assigned to audio-only and mixed video-audio?

Validity of the findings

- Line 197 "homogeneity was ensured". There are actually substantial differences between the 2 groups, eg % male 58.3% vs 33.3%. Your p values were all >0.05 only because of small sample size. Also you compared 4 characteristics, when there are many more characteristics that influence performance of CPR. I don't think you should so assertively say that homogeneity was ensured.
- What were the baseline CPR training of the participants? How about separately in the 2 arms. Prior training is probably one of the most important determinent of DACPR quality
- Line 413 "This study confirmed the difference". I don't think your study design allows such a strong conclusion. Suggest soften this assertion
- Generalizability of findings severely limited because there is no current system that uses AED-drones.

Reviewer 2 ·

Basic reporting

The aim of this study was compare first responders’ cardiopulmonary resuscitation (CPR) performance when a dispatcher provides audio instructions only and when both audio and video instructions are given.

The manuscript is written in a clear English, unambiguous, technically correct text. It is conform to professional standards of courtesy and expression. The manuscript include sufficient introduction and background to demonstrate how this article fits into this field of knowledge. The authors referenced relevant literature.

The article conform to an acceptable structure and format. The figures and tables of the article are relevant to the content of the manuscript.

Experimental design

The authors clearly define the aim anda research question. The manuscript authors identified the knowledge gap and statements were made as to how the study contributes to filling this gap.
The research have been conducted rigorously and to a high technical standard. The investigation have been conducted in conformity with the prevailing ethical standards in the field.
Methods have been described with sufficient information to be reproducible by another investigator

Validity of the findings

No comment

Reviewer 3 ·

Basic reporting

Overall hard work and good intentions and design. 1) not clear on methods of how Audio-visual coaching was accomplished (simple hand held cell phone by rescuer with both methods, or special/extra setup for drone/AV coaching 2) terminology of "Stop time" is atypical (usually named chest compression fraction 3) The authors report simple comparisons between groups, but neither group achieved good quality of CPR (slow CC rates, poor # of ventilations)...there were statistical improvements without clinical significance. 4) the qualitative findings were interesting 5) unclear the content or training of the dispatchers and coaches (e.g. good audio coaching beats bad audio/visual coaching and visa versa)

Experimental design

MIxed methods:
Improvement to be made by comparing the two groups for COMPLIANCE to current guidelines (not simple comparison on one vs other)...comparison of 47 CC/min vs 63 CC/min is stat sigif different, but dismal in both groups....same for 3.5 vs 5 ventilation/min.
Both had good CCF (83% vs 90%), but stat signif different.

Validity of the findings

see above: caution for statistical differences vs clinically important differences and the lack of analysis of differences without a distinction. Going for terrible to bad cpr compression rates and ventilation rates may be statistically different but not important to patient outcome.

Good start on qualitative findings. intreresting results.

---

## Round 0.2 · accepted · Accept

The authors addressed the reviewers' concerns and substantially improved the content of the manuscript.

So, based on my own assessment as an academic editor, no further revisions are required and the manuscript can be accepted in its current form.

Reviewer 1 ·

Basic reporting

My previous comments have been satisfactorily addressed.

Experimental design

My previous comments have been satisfactorily addressed.

Validity of the findings

My previous comments have been satisfactorily addressed.

Additional comments

My previous comments have been satisfactorily addressed.

Reviewer 2 ·

Basic reporting

I do not feel the need to re-review

Experimental design

.

Validity of the findings

.

Additional comments

.